:✺: PLOS | ONE

# Incidence, risk factors and healthcare costs of central line-associated nosocomial bloodstream infections in hematologic and oncologic patients

Claas Baier[1], Lena Linke[1], Matthias Eder[2], Frank Schwab[3], Iris Freya Chaberny[4], Ralf-Peter Vonberg[1], Ella Ebadi[1]*

1 Institute for Medical Microbiology and Hospital Epidemiology, Hannover Medical School, Hannover, Germany, 2 Department of Hematology, Hemostasis, Oncology, and Stem Cell Transplantation, Hannover Medical School, Hannover, Germany, 3 Institute of Hygiene and Environmental Medicine, Charité - University Medicine, Berlin, Germany, 4 Institute of Hygiene, Hospital Epidemiology and Environmental Medicine, Leipzig University Hospital, Leipzig, Germany

* ebadi.ella@mh-hannover.de

**Data Availability Statement:** Patient data used in this study is confidential according to the German data privacy act, the ethics committee and the data

## Abstract

Non-implanted central vascular catheters (CVC) are frequently required for therapy in hospitalized patients with hematological malignancies or solid tumors. However, CVCs may represent a source for bloodstream infections (central line-associated bloodstream infections, CLABSI) and, thus, may increase morbidity and mortality of these patients. A retrospective cohort study over 3 years was performed. Risk factors were determined and evaluated by a multivariable logistic regression analysis. Healthcare costs of CLABSI were analyzed in a matched case-control study. In total 610 patients got included with a CLABSI incidence of 10.6 cases per 1,000 CVC days. The use of more than one CVC per case, CVC insertion for conditioning for stem cell transplantation, acute myeloid leukemia, leukocytopenia ($\leq$ 1000/ µL), carbapenem therapy and pulmonary diseases were independent risk factors for CLABSI. Hospital costs directly attributed to the onset of CLABSI were 8,810 € per case. CLABSI had a significant impact on the overall healthcare costs. Knowledge about risk factors and infection control measures for CLABSI prevention is crucial for best clinical practice.

## Introduction

Bloodstream infections (BSI) are quite common in patients with hemato-oncologic diseases as they are often severely immunocompromised due to the underlying disease, to antineoplastic therapy, and/or to hematopoietic stem cell transplantation (HSCT) [1–3]. Nosocomial BSIs are often associated with the usage of some kind of invasive device entering the venous blood system such as central vascular catheters (CVC) and are then called central line-associated bloodstream infection (CLABSI) [4]. One meta-analysis on mostly ICU patients suggests that

protection commissioner of the Hannover Medical School. Patient related data such as ward of admission, age, sex, underlying disease or length of stay are indirect identifiers and might enable to track back the identity of a patient. To protect patient confidentiality and participant's privacy, data used for this study can be obtained in anonymous and condensed form only according to the data privacy act. Interested researchers may contact the data protection commissioner of the Hannover Medical School (datenschutz@mh-hannover.de) and the corresponding author (ebadi.ella@mh-hannover.de) to get access to anonymized data, approved by the data protection commissioner of the Hannover Medical School.

**Funding:** This research did not receive any specific grant from funding agencies in the public, commercial, or not-for-profit sectors.

**Competing interests:** The authors declare that they have no competing interests.

CLABSI deteriorates the clinical course of patients and contributes to overall mortality [5]. Thus, CLABSI prevention in heavily immunocompromised patients is considered crucial including knowledge of risk factors and the implementation of multimodal strategies and infection control concepts [6].

Besides the clinical consequences, CLABSI and other healthcare-associated infections may increase costs for both hospitals and healthcare systems [7, 8].This study therefore provides data on the incidence of primary nosocomial CLABSI in hematologic and oncologic patients, as well as on contributing risk factors and the financial burden of CLABSI.

## Materials and methods

### Setting

The Department of Hematology, Hemostasis, Oncology and Stem Cell Transplantation at the Hannover Medical School is a tertiary referral center for adults with hematological malignancies, benign diseases of hematopoiesis and solid tumors. There were two wards, one 20-bed ward mainly for patients undergoing hematopoietic stem cell transplantation and one 28-bed ward for patients receiving antineoplastic therapy mainly for all kinds of leukemia and lymphoma. CVCs got inserted exclusively by physicians of the local anesthesiology department or the intensive care unit according to published standards [9].

### Study design

Patients with and without CLABSI were included in a retrospective cohort study over a 3 year time period. Criteria for inclusion were as follows: i) age ≥18 years, ii) presence of a non-implanted CVC for at least 48 hours and iii) informed consent. Patients with secondary BSI were excluded. This is a retrospective, anonymous analysis of clinical and epidemiological data only, thus an ethical approval was not required.

### Definition of CLABSI

CLABSI was considered as nosocomially acquired if its onset took place at least 2 days after admission to one of the above mentioned wards. CLABSI was considered as a primary infection if there were no other clinical signs or symptoms for another infectious focus. Centers of Disease Control and Prevention (CDC) criteria (National Healthcare Safety Network) for primary (laboratory confirmed) BSI were applied [10]. CVC-association of the infection was defined as follows: i) the catheter was in place for at least 48 hours prior to onset of sepsis, and/or ii) there was microbiologic growth (bacteria and/or fungi) of at least 15 colony forming units (CFU) on the CVC tip identical to a positive blood culture sample, and/or iii) the difference in time to positivity between a central and a peripheral drawn blood culture was more than 2 hours. Patient days and CVC days were collected from the patient records.

### Statistical analysis

**CLABSI incidence and risk factors.** CLABSI incidence rates were calculated as the number of CLABSI cases per 1,000 CVC days. In the univariable analysis demographic and clinical characteristics of CLABSI patients and non-CLABSI patients were compared and differences were tested using relative risks (RR) with 95%-confidence intervals. In addition, a multivariable logistic regression analysis with stepwise variable selection forward was performed with the outcome CLABSI. The significance levels for entering a variable into the model was p = 0.05 and for removing a variable from the model was p = 0.06, respectively. For the risk

factor analysis in the CLABSI patients the time at risk (from admission to CLABSI) and in the non-CLABSI patients the time period from admission to discharge/death were considered.

**Costs and reimbursements.** For cost analysis, a matched case-control study was carried out. Patients with CLASBI got matched in a ratio of 1:2 to patients without CLASBI. The matching criteria for controls in the cost analysis were as follows: i) age within a 5 year range, ii) identical German diagnosis-related group (G-DRG) including split-difference, iii) admission in the same year and iv) hospital length of stay at least as long as the length until onset of CLABSI in the corresponding case. Data on costs and reimbursements for cases and controls were provided by the financial and controlling department on basis of G-DRGs. Costs were considered as costs of the whole hospital stay (aggregated costs from admission to discharge).

We calculated the median and interquartile range for cases and controls to compare costs and reimbursements. Differences between the groups were checked by the unpaired Wilcoxon rank sum test. Attributable costs and loss due to CLABSI were determined by matched pairs. The difference between CLABSI cases and non-CLABSI controls was tested by the paired Wilcoxon rank sum test. Additionally, to analyze independent factors which affect the costs a multivariable analysis were performed. Costs were log transformed to achieve normal distribution and a multivariable linear regression model was calculated by stepwise forward variable selection with $p = 0.05$ for entering a variable in the model and $p = 0.055$ for removing a variable. The regression coefficients were converted to the measures of effect using an exponential transformation and referred to as the multiplicative effect (ME) of patient characteristics.

Unless otherwise stated, p-values less than 0.05 were considered significant. All analyses were performed using SPSS (IBM SPSS statistics, Somers, NY, USA) and SAS 9.4 (SAS Institute, Cary, NC, USA).

## Results

### CLABSI incidence

A total of 610 cases met the inclusion criteria. The mean age was 47 years and 375 were male. The three most prevalent diseases were acute myeloid leukemia (n = 229), Non-Hodgkin lymphoma (n = 197) and acute lymphoblastic leukemia (n = 73). There were 172 patients, who underwent hematopoietic stem cell/bone marrow transplantation. The mean length of stay in the entire patient cohort was 26 days. Altogether 680 CVCs were inserted in these 610 cases adding up to 10,454 CVC days. Mean CVC usage time per case was 17 days (range 2–117). Most central lines were inserted in the jugular vein (n = 514), followed by subclavian vein (n = 159), basilic vein (n = 4) and femoral vein (n = 3). The majority of the catheters were impregnated with chlorhexidine/silver sulfadiazine (n = 652). There were 111 cases with primary nosocomial CLABSI (prevalence: 18.2%) resulting in a CLABSI incidence rate of 10.6 cases per 1,000 CVC days. Table 1 shows the corresponding causative microorganisms. Patients with CLABSI had a longer mean overall hospital stay compared to non-CLABSI cases (47 vs. 22 days, p<0.001). Mortality in the entire study population was 4% (26/610), however there was no significant difference between patients with and without CLABSI (7% vs. 4%; p = 0.115).

### Risk factors

Table 2 shows the results of the univariable risk factor analysis (selection of items). All items can be found in S1 Table. In the multivariable logistic regression analysis, the use of more than one CVC per case, CVC insertion for conditioning, acute myeloid leukemia, leukocytopenia ($\leq 1000/\mu L$), carbapenem therapy and pulmonary diseases were independent risk factors for

**Table 1. Pathogens causing central line-associated bloodstream infections (CLABSI).**

| Pathogen | Number (percentages) of CLABSI cases (n = 111) |
|---|---|
| Coagulase negative staphylococci | 81 (73.0) |
| *Staphylococcus epidermidis* | 72 (64,9) |
| *Staphylococcus haemolyticus* | 9 (8,1) |
| *Enterococcus spp.* | 16 (14.4) |
| *Enterococcus faecalis* | 11 (9,9) |
| *Enterococcus faecium* | 5 (4,5) |
| *Escherichia coli* | 6 (5,4) |
| *Corynebacterium amycolatum* | 3 (2,7) |
| *Streptococcus parasanguis* | 2 (1,8) |
| *Fusobacterium nucleatum* | 2 (1,8) |
| Other | 12 (10,8) |

The total number of pathogens exceeds the total number of patients (n = 111) as some CLABSI episodes were caused by more than one pathogen.

the onset of CLABSI. Independent protective factors were transfusion of erythrocytes, use of the subclavian vein as CVC insertion site and glycopeptide therapy (Table 3).

## Costs

Matching was possible for 79 cases and 158 controls. Cases and controls showed no significant difference in terms of age, gender and mortality (S2 Table). Significant differences were found between cases and controls with respect to overall length of stay and length of central line utilization. The total hospital costs for a patient with CLABSI significantly exceeded the costs for patients without CLABSI (Table 4). Hospital costs directly attributed to the onset of CLABSI as calculated by the difference in costs of matched pairs were 8,810 €. Comparison of single cost items between CLABSI cases and non-CLABSI controls showed significant differences e.g. for pharmaceuticals (2,117 € vs. 1541 €; p = 0.001), nurses (7,083 € vs. 6,061 €; p = 0.003) and medical products (3,451 € vs. 2,838 €; p = 0.02).

The multivariable analysis revealed factors affecting costs independently. It was confirmed that CLABSI significantly increased the hospital costs (ME = 1.4; p<0.001). Moreover, costs were independently increased by chemotherapy >5 days (ME = 2.4; p<0.001). In contrast acute myeloid leukemia (ME = 0.63; p<0.001), age >50 years (ME = 0.75; p = 0.001), and malignancy of the testicles (ME = 0.55; p = 0.03) were associated with lower hospital costs.

## Discussion

In this study, there was a CLABSI incidence rate of 10.6 per 1,000 CVC days and a CLABSI prevalence of 18.2%. These results are in line with other reports. For example, Tarpatzi et al. found a comparable incidence rate of 11.5 for (mainly) non-implanted CVCs in a tertiary Greek hospital including patients with hematological malignancy [11]. Looking at patients with cancer and several types of CVCs, Mollee et al. found a rather low overall CLABSI incidence rate of only 2.5. However, in a subgroup analysis of patients with aggressive hematological neoplasia, this incidence rate raised back to 17.3 for non-tunneled CVCs [12]. Luft et al. found a CLABSI incidence rate of 5.6 in hematological patients [13] while others report a general BSI rate in the pre-engraftment phase of 21% in hematologic patients undergoing HSCT [14]. These variations in the incidence and prevalence rates for BSI may be due to heterogeneity in the characteristics of hematologic patient populations including but not limited to the

**Table 2. Univariable analysis of factors (selection) influencing the risk of central line-associated bloodstream infections (CLABSI) significantly.**

| Influencing factors | No. of patients with CLABSI | No. of patients without CLABSI | RR* | 95%CI** | p-value*** |
|---|---|---|---|---|---|
| **total** | 111 | 499 | | | |
| **Risk factor** | | | | | |
| Age >50 years | 64 | 224 | 1.52 | 1.08–2.14 | 0.016 |
| Acute myeloid leukemia | 72 | 157 | 3.07 | 2.16–4.37 | <0.001 |
| Cardiac disease (comorbidity) | 37 | 110 | 1.57 | 1.11–2.23 | 0.014 |
| Body mass index >30 kg/m$^2$ | 21 | 59 | 1.55 | 1.02–2.34 | 0.061 |
| Carbapenem therapy | 32 | 75 | 1.90 | 1.34–2.71 | 0.001 |
| Aminoglycoside therapy | 41 | 61 | 2.92 | 2.12–4.02 | <0.001 |
| Hematopoietic stem cell transplantation | 47 | 125 | 1.87 | 1.34–2.61 | <0.001 |
| Allogenic hematopoietic stem cell/bone marrow transplantation | 32 | 70 | 2.02 | 1.42–2.87 | <0.001 |
| Leukocytopenia <1,000/µL | 109 | 279 | 31.18 | 7.78–125.05 | <0.001 |
| Anemia | 108 | 336 | 13.46 | 4.33–41.80 | <0.001 |
| Thrombocytopenia | 109 | 326 | 21.93 | 5.48–87.81 | <0.001 |
| >1 CVC inserted | 38 | 28 | 4.29 | 3.19–5.78 | <0.001 |
| CVC insertion for conditioning phase | 44 | 67 | 2.95 | 2.14–4.07 | <0.001 |
| Jugular vein insertion as CVC insertion site | 95 | 372 | 1.82 | 1.11–2.98 | 0.013 |
| **Protective factor** | | | | | |
| Non Hodgkin Lymphoma | 18 | 179 | 0.41 | 0.25–0.65 | <0.001 |
| Transfusion of erythrocytes | 27 | 286 | 0.31 | 0.20–0.46 | <0.001 |
| Subclavian vein as CVC insertion site | 15 | 132 | 0.49 | 0.30–0.82 | 0.003 |
| Length of CVC usage <8 days | 13 | 203 | 0.24 | 0.14–0.42 | <0.001 |

*Risk ratio.

** 95%-confidence interval.

***Fisher Exact Test (2-tailed p).

**Table 3. Independent factors significantly influencing the risk of central line-associated bloodstream infections (CLABSI) (multivariable logistic regression analysis).**

| Factor | OR* | 95%CI** | p-value |
|---|---|---|---|
| **Risk factors** | | | |
| Leukocytopenia <1,000/µL | 69.77 | 15.76–308.86 | <0.001 |
| >1 CVC inserted | 7.08 | 2.95–17 | <0.001 |
| Carbapenem therapy | 6.02 | 2.29–15.83 | <0.001 |
| Pulmonary diseases | 3.17 | 1.32–7.62 | 0.010 |
| Acute myeloid leukemia | 2.72 | 1.43–5.17 | 0.002 |
| CVC insertion for conditioning phase | 2.07 | 1.04–4.1 | 0.037 |
| **Protective factors** | | | |
| Transfusion of erythrocytes | 0.04 | 0.02–0.08 | <0.001 |
| Glycopeptide therapy | 0.10 | 0.03–0.34 | <0.001 |
| Subclavian vein as CVC insertion site | 0.32 | 0.14–0.77 | 0.010 |

Multivariable logistic regression with the outcome CLABSI; variable selection stepwise forward: p = 0.05 for including a parameter in the model and p = 0.06 for removing a parameter. CVC = central venous catheter

*Odds ratio

**95%-confidence interval.

**Table 4. Costs and reimbursements (in Euro) for case patients with central line-associated bloodstream infection (CLABSI) and control patients without CLABSI in the matched case-control study (1:2)[\*].**

| | Case patients with CLABSI (n = 79) | Control patients without CLABSI (n = 158) | p-value |
|---|---|---|---|
| **Median costs** | 54,454 (25,634–112,697) | 48,965 (17,538–78,706) | 0.025[a] |
| **Median reimbursements** | 74,662 (18,331–82,801) | 74,662 (17,478–79,745) | 0.290[a] |
| **Median loss/profit** | -8,888 (-29.993–2,522) | 1,000 (-11,198–9,581) | <0.001[a] |
| **Median costs attributable to CLABSI** | 8,810 (-2,237–3,487) | | <0.001[b] |
| **Median loss attributable to CLABSI** | -8,171 (-29,090–2,396) | | <0.001[b] |

[\*]matching criteria for controls in the cost analysis were as follows: i) age within a 5 year range, ii) same German diagnosis-related group (G-DRG), iii) admission in the same year, iv) hospital length of stay at least as long as the length until onset of CLABSI in the corresponding case.

CLABSI = Central line-associated bloodstream infection

[a] Wilcoxon rank sum test

[b] Wilcoxon rank sum test for paired samples.

presence of individual risk factors, distribution of underlying diseases, and the proportion of neutropenic patients.

Furthermore, the exact definition for primary BSI in general and CLABSI in particular may differ between studies. For better comparison, we recommend to rely here on the well-established CDC criteria for primary BSI and to consider CLABSIs only as such if occurred at least 48 hours after insertion of the CVC.

Biofilm is a well-known and pivotal problem when dealing with infections due to contaminated CVCs as demonstrated in a recent review by Ielapi et al. [15]. In fact, preventive measures during insertion and thereafter are crucial to prevent biofilm formation and bacterial colonization of the catheter. These may include lock therapies or coated materials besides best nursing practices [15]. In order to address this topic in our definition, significant bacterial growth on the central line tip simultaneous to a positive blood culture and/or the difference in time to positivity between a central and peripheral blood culture of more than 2 hours was used in our study to support the thesis of the CVC being the actual origin of the infection [16].

Moreover, incidence rates may depend on the study type (prospective vs. retrospective approach). In prospectively planned studies, it is more likely that control measures such as a daily check of the insertion site are implemented [17], which in consequence raises the general awareness for CLABSI. A more standardized manner of monitoring BSI rates may overcome those shortcomings and thus should allow valid longitudinal performance evaluation. If such a surveillance takes places in a standardized mode specifically designed for hemato-oncologic patients [18], a cross sectional comparison with other participating institutions would be possible and may prove very beneficial.

In the multivariable risk factor analysis, a leukocytopenia proved to be an independent risk factor for CLABSI. This is in line with several other reports that found severe leukocytopenia and neutropenia being relevant risk factors for CLABSI development in adult and pediatric patients with different implanted and non-implanted CVCs [13, 19–21]. Leukopenia may correspond to the intensity of therapy and state of disease, in particular in acute leukemia. In contrast, Huang et al. recently did not show a significant association between a low absolute leukocyte count and the risk for BSI development in patients with multiple myeloma. However, their study did not focus on CLABSI in particular [22].

The usage of more than one central line was also an independent risk factor for CLABSI onset in our study. In a univariable analysis, Kelly et al. came up with similar results in pediatric oncology patients [23]. On the one hand, a further consecutive central line insertion

represents an additional potential entry mechanism for pathogens into the patient's blood flow; on the other hand, the need of more than one (consecutive) CVC could also be just a surrogate parameter for a more severe clinical course of disease, which in itself predisposes patients to all kinds of infections. Moreover, we observed that CVC change was often triggered by local or systemic signs and symptoms of infection (not necessarily fulfilling CLABSI criteria at that time point, but predisposing patients to manifest CLABSI later on). The same phenomenon might apply to carbapenem therapy, which was also determined as independent risk factors for CLABSI in our study. In fact, during the study period carbapenems were part of antibiotic therapy escalation schemes (persisting fever) and might therefore just indicate patients with a higher grade of severity of illness and, thus, increased likelihood of BSI. Acute myeloid leukemia was also an independent risk factor for CLABSI which is in line with findings by Mollee et al. [12]. Interestingly, a pre-existing pulmonary disease was an independent risk factor in our patient cohort as well. This was surprising to us and needs further investigation in future studies. To our surprise, the factors length of CVC usage and length of leukocytopenia did not influence the development of CLABSI in the multivariable analysis.

In our study, the insertion site did influence the risk for CLABSI in the multivariable analysis. The subclavian vein as insertion site was protective. In fact, there are some reviews and studies focusing on the CVC insertion site and its effect on outcome parameters such as infection or mechanical complications. In a French study from 2015 including 10 intensive care units, the choice of the subclavian vein for CVC insertion was protective with respect to the onset of BSI [24]. Other studies found the femoral vein to be a risk factor for infectious complications such as CLABSI [25]. In our study, only very few catheters were inserted in the femoral vein so a reliable statement concerning risk of the femoral insertion site cannot be made. A review by Marik et al. from 2012 showed no differences in the CLASBI rate between the three typical insertion sites (femoral, jugular and subclavian) [26] and another review concluded that an increased risk of CLABSI for the femoral insertion site is arguable [27].

Unexpectedly, erythrocyte transfusions also were a protective factor in our study cohort. This is in contrast, to previous findings of others [28, 29]. We cannot provide a plausible explanation for this observation yet. In our view, these findings need further evaluation in large-scale populations.

In our cohort, patients with CLABSI had a prolonged mean hospital stay. However, based on our clinical experience, especially with CLABSI caused by coagulase negative s*taphylococci*, we doubt that CLABSI represents the true cause for this finding in our cohort. In fact, evaluating independent risk factors for a prolonged hospital stay in hematologic patients was beyond the scope of this study. So after all this issue remains unclear. A potential bias concerning the distribution of underlying malignancies or procedures performed in the CLABSI and non-CLABSI cohort appears to be a more likely explanation; for instance the proportion of patients undergoing HSCT was larger in the CLABSI cohort (47/111 = 42.3%) than it was in control patients without CLABSI (125/499 = 25.1%) as shown in Table 2.

Our study revealed attributable median costs of 8,810 € for each CLABSI case. These additional costs result in an attributable median loss of 8,171 € per CLABSI case as costs exceeded reimbursements (Table 4). Other studies have also addressed the costs of nosocomial BSI and CLABSI before and found a relevant increase of costs of several thousands of Dollars or Euros (e.g. about 10,000–70,000) per each case in comparison to controls [30–34].

Multivariate analysis confirmed CLABSI being an independent risk factor increasing costs (ME = 1.4). An application of chemotherapy for >5 days also increased costs independently. In contrast, we found that acute myeloid leukemia (AML), a malignancy of the testicles, and age of >50 years decreased costs in our study. Advanced age was associated with reduced costs from nosocomial infections in other studies, too, but this effect may be due to increased

mortality and, thus, a reduced length of stay of affected patients [35]. Moreover, especially older patients receive less often intensive chemotherapies, which might have contributed to these findings. The similar findings for AML and malignancy of the testicles as underlying diseases remain yet unclear. These topics should be addressed in further investigations, preferably in larger patient cohorts.

Our study has some limitations: 1) All patients in our cohort were cared for in one facility (Hannover Medical School in Germany) from 2004 to 2006. The results of such a retrospective single-centre study might therefore not apply to other hospitals and settings. 2) Moreover, a frequent challenge when comparing CLABSI surveillance data (incidence, risk factors, and costs) is the lack of a standardized CLABSI definition in particular. We used rather heterogeneous definitions for CLABSI although at least based on the well-known CDC definitions for primary laboratory confirmed BSI. The amendment of the subcategory "mucosal barrier injury caused BSI" was published by the CDC (National Healthcare Safety Network) [36] later on after collection of our data was finished. However, this category is of relevance especially for patients with hematological malignancies who often develop severe mucosal damages (mucositis) during their clinical course, for instance caused by a Graft-versus-Host disease or during conditioning. So, even if a central line is present and the CDC criteria for primary CLABSI are fulfilled, BSI in these cases may rather origin from the injured gut or oral mucosa rather than from the CVC. As a consequence, this infection usually presents with typical microorganisms from the gut or oral cavity such as *Escherichia coli*, *enterococci* or *streptococci* [37] and "traditional" infection control strategies proposed to prevent CLABSI will fail in these subgroups [38]. However, in our study period coagulase-negative *Staphylococci* dominated the spectrum of pathogens and, thus, we believe that the bias of false classification of mucositis-induced BSI is rather small in our data. Nevertheless, in future studies regarding risk factors, costs and incidence of CLABSI this novel subgroup should be carefully considered. 3) We did not implement a general morbidity scoring system which combines clinical items to assess its impact on the CLABSI risk as done by others [39]. However, we assume that the independent risk factors "carbapenem therapy" and ">1 CVC inserted" somehow reflect such a general predisposition in our study and could function as a surrogate parameter. 4) Only a rather small sample size was available for cost analysis due to the strict matching criteria applied. However, in our view such strict criteria are necessary in order to warrant a comparable control group and to minimize bias. One major criterion for matching was an identical G-DRG code. However, differences in the therapy regimes applied might not always be adequately depicted by DRG codes. 5) Finally, we did not differentiate costs before and after the onset of CLABSI, however we believe that it is very likely that the increased costs occur mainly after the onset of CLABSI.

## Conclusion

This data on the clinical and economic impact of CLABSI underline the need for proper adherence to appropriate and recommended infection control measures. This is especially important when caring for patients in the phase of leukocytopenia. Reduction of infections will significantly reduce morbidity, mortality and healthcare costs

## Supporting information

**S1 Table. All items included in the univariable analysis.** For patients with central line-associated bloodstream infections (CLABSI) the time at risk (admission to onset of CLABSI) is used, for the non-CLABSI patients the time from admission to discharge or death is used. (DOCX)

**S2 Table. Cost analysis.** Selected characteristics of the case patients and the control patients. For patients with central line-associated bloodstream infections (CLABSI) the time at risk (admission to onset of CLABSI) is used, for the non-CLABSI patients the time from admission to discharge or death is used.
(DOCX)

## Acknowledgments

We would like to thank the financial and controlling department of the Hannover Medical School for providing financial data.

## Author Contributions

**Conceptualization:** Iris Freya Chaberny, Ella Ebadi.

**Data curation:** Claas Baier, Lena Linke, Ella Ebadi.

**Formal analysis:** Lena Linke, Frank Schwab.

**Investigation:** Lena Linke, Ella Ebadi.

**Methodology:** Iris Freya Chaberny, Ella Ebadi.

**Supervision:** Matthias Eder, Iris Freya Chaberny, Ella Ebadi.

**Validation:** Claas Baier, Ella Ebadi.

**Writing – original draft:** Claas Baier, Ralf-Peter Vonberg, Ella Ebadi.

**Writing – review & editing:** Claas Baier, Lena Linke, Matthias Eder, Frank Schwab, Iris Freya Chaberny, Ralf-Peter Vonberg, Ella Ebadi.

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
