## [Decision Letter · Decision Letter 0]

3 Nov 2019

PONE-D-19-22966

Incidence, risk factors and healthcare costs of central line-associated nosocomial bloodstream infections in hematologic and oncologic patients

PLOS ONE

Dear Dr. Ebadi,

Thank you for submitting your manuscript to PLOS ONE. After careful consideration, we feel that it has merit but does not fully meet PLOS ONE’s publication criteria as it currently stands. Therefore, we invite you to submit a revised version of the manuscript that addresses the points raised during the review process.

The paper is potentially interesting and will be reconsidered provided you are willing to revise it according to the reviewer's suggestions.

We would appreciate receiving your revised manuscript by Dec 18 2019 11:59PM. To enhance the reproducibility of your results, we recommend that if applicable you deposit your laboratory protocols in protocols.io, where a protocol can be assigned its own identifier (DOI) such that it can be cited independently in the future. For instructions see: http://journals.plos.org/plosone/s/submission-guidelines#loc-laboratory-protocols

We look forward to receiving your revised manuscript.

Kind regards,

Raffaele Serra, M.D., Ph.D

Academic Editor

PLOS ONE

Journal Requirements:

Additional Editor Comments:

The manuscript is interesting. Some extra work is needed in order to improve the discussion section.

Reviewers' comments:

Reviewer's Responses to Questions

**Comments to the Author**

1. Is the manuscript technically sound, and do the data support the conclusions?

Reviewer #1: Yes

2. Has the statistical analysis been performed appropriately and rigorously? 

Reviewer #1: Yes

3. Have the authors made all data underlying the findings in their manuscript fully available?

Reviewer #1: Yes

4. Is the manuscript presented in an intelligible fashion and written in standard English?

Reviewer #1: Yes

5. Review Comments to the Author

Reviewer #1: Good and timely study on an important issue. This study there provides information on the incidence of primary nosocomial CLABSI in hematologic and oncologic patients, discussion also risk factors and the financial burden of CLABSI.

I suggest to improve the discussion section outlining the importance of biofilm, that is a pivotal factor in CLABSI as current and recent studies well show. For this purpose please see, cite and comment the recent reviews by Ielapi N. et al. The Role of Biofilm In Central Venous Catheter Related Bloodstream Infections: Evidence Based Nursing And Review of The Literature. Rev Recent Clin Trials. 2019 Oct 18. doi: 10.2174/1574887114666191018144739. [Epub ahead of print].

6. PLOS authors have the option to publish the peer review history of their article (what does this mean?). If published, this will include your full peer review and any attached files.

Reviewer #1: No

---

## [Author Response · Author response to Decision Letter 0]

11 Dec 2019

Dear Raffaele Serra (M.D., Ph.D.), 

Thank you very much for the reply to our manuscript PONE-D-19-22966. 

We strongly appreciate the comments of the reviewer. They were very constructive and helped improving the manuscript considerably. We addressed all of the reviewer’s points (see below) and amended the manuscript accordingly. The additions and changes regarding the content are indicated in the revised manuscript in yellow colour. The changes regarding PLOS ONE's style requirements are indicated with tracked changes (Microsoft Word) in the revised manuscript. 

Moreover, we would like to thank the scientific editor for handling the manuscript and giving us further valuable advice for improving it. We addressed the editor’s points as well in the revised manuscript. 

We hope very much that the manuscript is acceptable in its revised form for publication and are looking forward to your decision.

Thank you again.

Sincerely,

Ella Ebadi 

(On behalf of the authors)

Scientific editor’s comments:

Editor: We note that you have included the phrase “data not shown” in your manuscript. Unfortunately, this does not meet our data sharing requirements. PLOS does not permit references to inaccessible data. We require that authors provide all relevant data within the paper, Supporting Information files, or in an acceptable, public repository. Please add a citation to support this phrase or upload the data that corresponds with these findings to a stable repository (such as Figshare or Dryad) and provide and URLs, DOIs, or accession numbers that may be used to access these data. Or, if the data are not a core part of the research being presented in your study, we ask that you remove the phrase that refers to these data.

Authors: We added a supplementary table 2 (S2 table) to the revised manuscript. We therefore deleted the phrase “data not shown” and now reference S2 table in the revised manuscript (page 10, line 176). 

Editor: We note that you have indicated that data from this study are available upon request. PLOS only allows data to be available upon request if there are legal or ethical restrictions on sharing data publicly. For information on unacceptable data access restrictions, please see http://journals.plos.org/plosone/s/data-availability#loc-unacceptable-data-access-restrictions.

Authors: We indicated that data from this study are available upon request. There are some legal restrictions for data access in our case, which are specified as follows: 

“Patient data used in this study is confidential according to the German data privacy act, the ethics committee and the data protection commissioner of the Hannover Medical School. Patient related data such as ward of admission, age, sex, underlying disease or length of stay are indirect identifiers and might enable to track back the identity of a patient. To protect patient confidentiality and participant’s privacy, data used for this study can be obtained in anonymous and condensed form only according to the data privacy act. Interested researchers may contact ebadi.ella@mh-hannover.de to get access to anonymized data, approved by the data protection commissioner of the Hannover Medical School.“

Editor: The manuscript is interesting. Some extra work is needed in order to improve the discussion section.

Authors: We thank the editor for this evaluation of our manuscript. We fully agree that the discussion section can be improved by taking further aspects into account. As suggested by the reviewer, we would like to additionally address the aspect of biofilm formation on central venous catheters in the revised manuscript (see page 10, line 219-223). We hope the scientific editor agrees with this addition. 

Reviewer’s comment:

Reviewer: Good and timely study on an important issue. This study there provides information on the incidence of primary nosocomial CLABSI in hematologic and oncologic patients, discussion also risk factors and the financial burden of CLABSI.

I suggest to improve the discussion section outlining the importance of biofilm, that is a pivotal factor in CLABSI as current and recent studies well show. For this purpose please see, cite and comment the recent reviews by Ielapi N. et al. The Role of Biofilm In Central Venous Catheter Related Bloodstream Infections: Evidence Based Nursing And Review of The Literature. Rev Recent Clin Trials. 2019 Oct 18. doi: 10.2174/1574887114666191018144739. [Epub ahead of print].

Authors: We fully agree with the reviewer that biofilm formation is pivotal in the pathogenesis of CLABSI. The recent publication mentioned by the reviewer is very interesting and helps to better understand this aspect. It adds an important point to the discussion section of the revised manuscript. As suggested by the reviewer we now cite and comment on the publication by Ielapi at al. in the discussion section of the revised manuscript (see page 10, line 219-223).

---

## [Decision Letter · Decision Letter 1]

30 Dec 2019

Incidence, risk factors and healthcare costs of central line-associated nosocomial bloodstream infections in hematologic and oncologic patients

PONE-D-19-22966R1

Dear Dr. Ebadi,

We are pleased to inform you that your manuscript has been judged scientifically suitable for publication and will be formally accepted for publication once it complies with all outstanding technical requirements.

With kind regards,

Prof. Raffaele Serra, M.D., Ph.D

Academic Editor

PLOS ONE

Additional Editor Comments (optional):

amended manuscript is acceptable

Reviewers' comments:

Reviewer's Responses to Questions

**Comments to the Author**

1. If the authors have adequately addressed your comments raised in a previous round of review and you feel that this manuscript is now acceptable for publication, you may indicate that here to bypass the “Comments to the Author” section, enter your conflict of interest statement in the “Confidential to Editor” section, and submit your "Accept" recommendation.

Reviewer #1: All comments have been addressed

2. Is the manuscript technically sound, and do the data support the conclusions?

Reviewer #1: Yes

3. Has the statistical analysis been performed appropriately and rigorously? 

Reviewer #1: Yes

4. Have the authors made all data underlying the findings in their manuscript fully available?

Reviewer #1: Yes

5. Is the manuscript presented in an intelligible fashion and written in standard English?

Reviewer #1: Yes

6. Review Comments to the Author

Reviewer #1: The manuscript is now acceptable. The authors fully addressed my concerns. I think this article may be interesting for Plos One readership.

7. PLOS authors have the option to publish the peer review history of their article (what does this mean?). If published, this will include your full peer review and any attached files.

Reviewer #1: No

---

## [Editor Report · Acceptance letter]

17 Jan 2020

PONE-D-19-22966R1 

Incidence, risk factors and healthcare costs of central line-associated nosocomial bloodstream infections in hematologic and oncologic patients 

Dear Dr. Ebadi:

I am pleased to inform you that your manuscript has been deemed suitable for publication in PLOS ONE. Congratulations! Your manuscript is now with our production department. 

With kind regards,

on behalf of

Prof. Raffaele Serra 

Academic Editor

PLOS ONE